# Application of Machine Learning in Predicting Perioperative Outcomes in Patients with Cancer: A Narrative Review for Clinicians

Garry Brydges [1], Abhineet Uppal [2] and Vijaya Gottumukkala [3,*]

1 Division of Anesthesiology, Critical Care & Pain Medicine, The University of Texas at MD Anderson Cancer Center, Houston, TX 77030, USA; gbrydges@mdanderson.org
2 Department of Colon & Rectal Surgery, The University of Texas at MD Anderson Cancer Center, Houston, TX 77030, USA; auppal1@mdanderson.org
3 Department of Anesthesiology & Perioperative Medicine, The University of Texas at MD Anderson Cancer Center, 1400-Unit 409, Holcombe Blvd, Houston, TX 77030, USA
* Correspondence: vgottumukkala@mdanderson.org; Tel.: +1-713-794-1398

**Abstract:** This narrative review explores the utilization of machine learning (ML) and artificial intelligence (AI) models to enhance perioperative cancer care. ML and AI models offer significant potential to improve perioperative cancer care by predicting outcomes and supporting clinical decision-making. Tailored for perioperative professionals including anesthesiologists, surgeons, critical care physicians, nurse anesthetists, and perioperative nurses, this review provides a comprehensive framework for the integration of ML and AI models to enhance patient care delivery throughout the perioperative continuum.

**Keywords:** artificial intelligence; machine learning; neural networks; perioperative outcomes





## 1. Introduction

Artificial intelligence (AI) is emerging as a pivotal tool in healthcare decision-making, with machine learning (ML) representing a significant subdomain that leverages advanced statistical techniques to facilitate autonomous learning through algorithms [1–3]. ML, rooted in computational science, analyzes data structures to discern patterns and extract insights, drawing from foundational principles in multivariate statistics [4,5]. The historical dichotomy between classical AI, focused on symbolic domains, and nascent neural network (NN) approaches led to contentious debates, particularly highlighted in Minsky and Papert's seminal work, "Perceptrons" [5–7]. Despite initial skepticism, the boundaries between AI and ML have gradually blurred over time, indicating a convergence trend [7].

In the complex landscape of healthcare delivery, the perioperative continuum of surgical practice stands out as a multifaceted domain encompassing pre-, intra-, and postoperative phases [4]. Integrating ML into surgical and perioperative care could yield substantial benefits, particularly in identifying factors—both modifiable and non-modifiable—across the continuum that significantly influence patient outcomes and healthcare costs. This article focuses on ML applications pertinent to predictive analytics in the perioperative continuum, emphasizing techniques such as exploratory data analysis (EDA), supervised learning (SL), and neural network learning (NNL). Addressing the challenges inherent in predicting perioperative outcomes using traditional methods, this review discusses the potential of ML integration in optimizing patient care delivery throughout the perioperative journey [2–4].

## 2. Methods

*Search Strategy and Source Selection*

This narrative review focuses on the integration of machine learning (ML) and neural network (NN) techniques in predictive analytics across the perioperative continuum.

Relevant Databases: PubMed, Embase, IEEEE Xplore, Medline, Scopus, and Web of Science.

Search Terms: "machine learning", "neural networks", "artificial intelligence", "perioperative care", "predictive analytics", "surgical outcomes", "oncological", "colorectal cancer", and "conceptual framework".

Inclusion/Exclusion Criteria: We included studies published in English between 2014 and 2024, focusing on ML and NN applications in the oncological perioperative settings. We excluded studies unrelated to perioperative oncological care, ML, NN, or AI, as well as non-English publications.

## 3. Understanding the Data

Data science has become increasingly important within healthcare organizations. Data science's value added is its ability to provide insights into managing resources across multiple departments (care units) while maximizing efficiency gains over time. These efficiency gains are primarily made through predictive analytic models built upon large volumes of structured information collected from various sources, including electronic health records (EHRs) [8]. Exploratory data analysis (EDA) provides an invaluable tool for these efforts by enabling analysts to find new ways to visualize complex datasets, which would otherwise require extensive manual effort to analyze and interpret accurately. As a result, EDA enables organizations to draw actionable conclusions much faster than with traditional methods. EDA allows for improved trending and timely decision-making processes across the perioperative practice [8–10].

EDA is a valuable tool for gaining insights into the structures of, patterns in, and trends in datasets. EDA involves visualizing, summarizing, and exploring relationships between variables to uncover patterns useful for decision-making and ML model building. By applying these methods, clinicians can identify potential trends or associations in their dataset that may not be apparent from traditional approaches [10,11]. EDA helps researchers better understand their datasets before they apply more formal analytical methods, such as ML regression or classification techniques. EDA could become a crucial component of clinical anesthesia and perioperative research as it provides valuable information that can be used to inform decisions related to patient care.

By examining existing datasets using visualization tools such as interactive heat maps, scatter plots, and boxplots, researchers can quickly identify potential areas where interventions could positively impact patient outcomes without prior knowledge of the significance of those interventions [2,4,8]. This type of analysis also allows clinicians to evaluate current treatment protocols by comparing them against historical results from previous studies conducted on similar populations so that they can make informed decisions regarding future changes in practice guidelines based on empirical evidence rather than anecdotal experience alone [2].

In addition to its use in clinical research settings, EDA has become increasingly popular in data science and analytics for discovering new knowledge about complex systems or processes by exploring large amounts of structured or unstructured digital information available on various platforms like social media networks or web-based applications [1,8]. Also important is the ability of EDA to efficiently inform data cleansing (e.g., identifying outliers and sparse datasets) and dimensionality reduction by identifying critical/essential variables. EDA is a crucial step taken prior to mathematical model selection and training. Through this approach, researchers can gain insight into clinician or patient behavior patterns through sentiment analysis, uncover hidden correlations between different types of medical treatments, and study how ML algorithms perform under certain conditions,

among the many other applications across industries ranging from healthcare delivery organizations to healthcare industry technologies [2–4].

## 4. ML Application in the Perioperative Continuum

ML enables improved predictive analytics throughout the perioperative continuum [12]. ML can incorporate a large number of independent variables from the various phases and domains of care to evaluate dependent variables such as postoperative ileus, surgical site infections, lengths of stay, readmission rates, and other postoperative outcomes [2,3,9,13–17]. While it is challenging to determine if variables are genuinely independent of historical data collection, many are interdependent. ML's value in predicting perioperative outcomes is its ability to train and test large amounts of complex data accurately, efficiently, and autonomously. The domain of artificial intelligence readily identifies patterns, trends, and abnormalities in real time for clinical decision-making support [8,14,16–19].

### 4.1. Understanding Machine Learning

ML is broken into four subdomains: (1) supervised learning, (2) unsupervised learning, (3) semi-supervised learning, and (4) reinforced learning (Table 1) [8].

**Table 1.** Subdomains of machine learning [1–8].

| ML Subdomain | Definition |
| --- | --- |
| Supervised learning (SL) | SL involves learning to predict future events by utilizing past events to perform dataset analysis (training) through inferred functions to make predictions (testing) regarding outcomes. The outcome (target) variable is known for predictions. ML algorithms enable error prediction and self-correction. Types of SL include (i) classification and (ii) regression [8]. Examples include training prediction models for risk indices and clinical research. |
| Unsupervised learning (UL) | UL learning involves algorithms that analyze and cluster unlabeled data according to hidden patterns or data groupings. Types of UL include (i) clustering, (ii) association rules, and (iii) dimensionality reduction [5]. Examples: medical imaging in pathology and radiology and clinical decision support. |
| Semi-supervised learning (SSL) | SSL learning combines supervised and unsupervised learning by utilizing a small sampling of labeled data plus a large amount of unlabeled data. Examples: speech recognition and text identification in the electronic health record [5]. |
| Reinforced learning (RL) | RL learning operates through sequential decision-making (trial and error) to maximize total reward through random trialing. Example: bioprosthetic devices [5]. |

### 4.2. Supervised Machine Learning

Supervised machine learning (SL) is a powerful tool that improves accuracy and efficiency in evaluating medical procedures on target (i.e., patient outcomes) variables [5]. SL uses algorithms to analyze data from patient records, medical images, or other sources to predict outcomes or treatments. SL models are trained on data where the outcomes (target variable) are historically known to make future outcome predictions. This technology has been applied to various aspects of anesthesiology and perioperative medicine, such as pre-operative assessment and intraoperative monitoring systems [1,13,14]. One example of SL being used in anesthesiology is using ML models for predicting post-surgical complications such as pulmonary compromise, cardiac arrest, or stroke [13,16–19]. SL models can also be used for hemodynamic optimization based on individual patients' demographic and physiological parameters, including blood pressure, stroke volume variation, and heart rate variability, among others [13,16]. Furthermore, SL can also be utilized by surgeons during laparoscopic and robotic procedures, providing real-time guidance through augmented reality displays, which help them to visualize anatomy more accurately than with traditional imaging techniques alone [4,14].

Overall, SL offers tremendous potential applications within anesthesiology and surgery that could improve clinical outcomes and operational efficiencies. SL's ability to process

enormous amounts of data quickly and its highly accurate predictive capabilities allow healthcare providers more insight into their patient's health status, thus enabling better decision-making processes throughout all stages of perioperative care [20,21]. However, SL's limitations or weakness occurs when historical target variables change over time due to newer technical innovations. As ML adoption develops rapidly, it will become increasingly integrated into daily practices across multiple perioperative specialty areas, improving healthcare quality worldwide [16].

Before selecting a machine learning model, the data must be clean, standardized, and representative of the variables used to make predictions. The data must be reviewed, and a decision must be made on handling missing values, outliers, duplication, and special characters. A range of techniques, from eliminating rows to replacement through imputation, can be performed during the data cleaning process [8,12–16].

The next step is model selection, choosing the best machine learning model for a given perioperative problem [14,20,21]. When making a model selection, factors include the type of problem (i.e., classification or continuous variables), the size of the dataset, and the desired model accuracy. Standard methods used in model selection incorporate a combination of cross-validation, feature selection, and hyperparameter tuning [14,21]. Also, the initial step should begin with simple model selection and evaluate the model performance metrics [21]. Model selection is a time-consuming process due to experimentation with different models and selection methods.

Next, the data are split into training and testing sets. Sometimes, a validation set is used in conjunction with the training set to enhance the ML model performance metrics [1,8,12,15]. The model is trained on the training data through an iterative process. The training model performance is evaluated for accuracy, precision, recall, and F1 scores. Training model performance is enhanced with hyperparameter tuning. Once the training model meets performance expectations, then the model moves into testing [8,12,15]. On occasion, the machine learning model can overfit, which is a model bias that learns the training data too well and cannot generalize to new data. The methods to counter overfitting are regularization techniques, changing to a smaller model, and cross-validation techniques, including a holdout set or early stopping techniques [8,12,15,18].

Once the training model meets satisfactory criteria, the model is either further optimized through hyperparameter-tuned training or evaluated on a validation set to enhance ML model performance metrics. Then, the model is evaluated for performance on the new testing data [8,18–21]. The goal is to achieve a high testing model performance to generalize any future unseen data adequately. Once the testing model performs well, the model is deployed to production, allowing others to use the model to make predictions, such as patient outcomes research. When ML models do not perform well, even after tuning, alternative ML models are trained and tested on the data. Using a well-constructed framework, new ML models are assessed for optimal generalizability and predictions and tested on the data (Figure 1) [18–21].

The conceptual framework incorporates data science trajectory, clinical trajectory, and research trajectory as the pillars for developing a perioperative cancer outcomes program. The data science trajectory collates the data sources into a standardized format to apply machine learning and neural network algorithms. The clinical trajectory utilizes data science to evaluate clinical care using elements of artificial intelligence. The research trajectory studies perioperative outcomes and develops clinical trials to further enhance clinical care pathway standardization.

While this framework offers a promising path towards improved perioperative care through prediction and decision support, limitations need to be acknowledged. The generalizability of ML models trained on specific datasets is a concern, and ensuring their effectiveness across diverse patient populations is crucial. Additionally, real-world implementation requires addressing data security and privacy issues. Integrating ML tools seamlessly into existing clinical workflows will necessitate collaboration between clinicians,

data scientists, and healthcare IT professionals. Overcoming these challenges will pave the way for the successful application of this framework in enhancing patient outcomes.

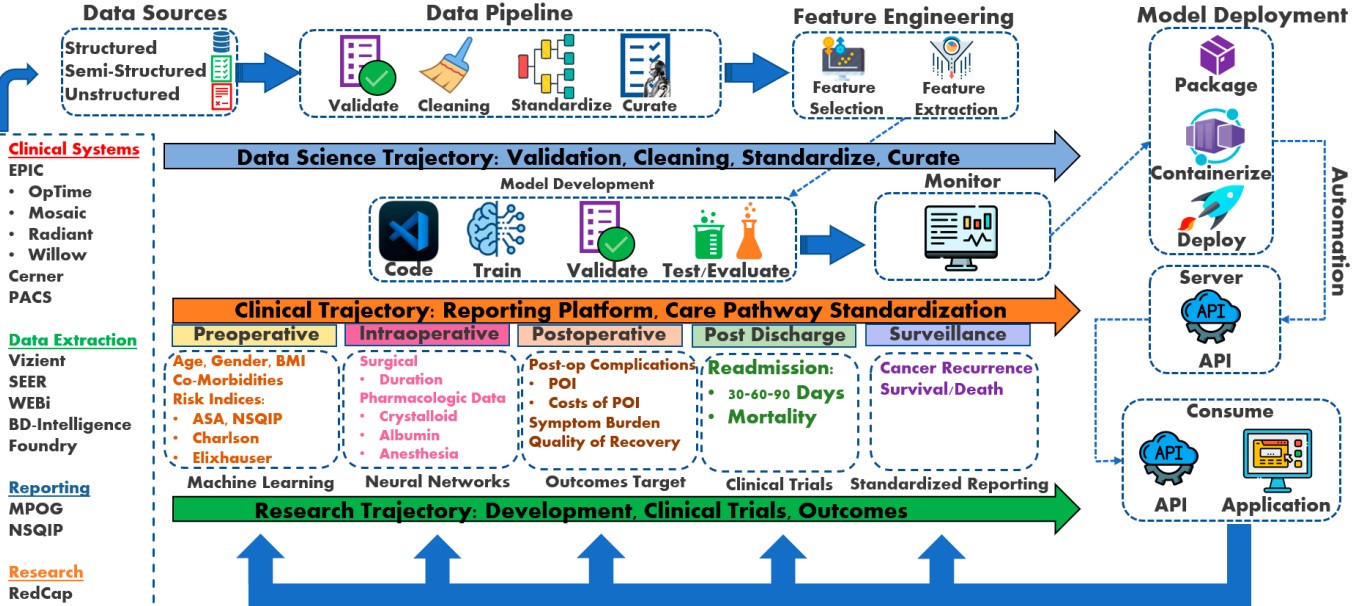

**Figure 1.** Perioperative Cancer Outcomes Group (PCOG) conceptual framework [8,21]. RedCap: Research Electronic Data Capture; NSQIP: National Surgical Quality Improvement Program; MPOG: Multicenter Perioperative Outcomes Group; WEBi: BusinessObjects Web Intelligence; SEER: Surveillance, Epidemiology, and End Results Program; PACS: Picture Archiving and Communication System; API: application programming interface; POI: postoperative ileus; AKI: acute kidney injury; RIOT: Return to Intended Oncologic Treatment.

In addressing the critical aspect of data collection for training machine learning (ML) models within the perioperative context, it is imperative to outline robust methodologies and ethical considerations. Data acquisition typically involves the aggregation of diverse sources, including electronic health records (EHRs), medical imaging, and patient monitoring systems, to construct comprehensive datasets representative of perioperative care. Ethical considerations loom prominently, as the utilization of patient data necessitates stringent adherence to privacy regulations, such as HIPAA in the United States, to safeguard patient confidentiality and autonomy. Furthermore, informed consent and transparent communication regarding data usage are paramount to uphold patient trust and mitigate concerns regarding data privacy breaches. Additionally, efforts to mitigate bias and ensure data integrity through rigorous preprocessing techniques, anonymization and deidentification strategies are indispensable. Collaborative partnerships between healthcare institutions and data science experts are essential to navigate the complex landscape of data acquisition while upholding ethical standards, ensuring the responsible and equitable use of patient data for advancing perioperative care through ML applications.

### 4.3. Forms of Supervised Machine Learning

The following models are introduced in order of relative complexity. While all machine learning models are robust mathematical models, model selection or complexity must match the complexity of the dataset. Selecting a simple ML model for complex data results in poor performance, such as accuracy, and in contrast, selecting an ML model that is too complex results in overfitting the dataset.

## 5. Classification Models: Logistic Regression, Classification (Decision) Tree

Invented by British statistician (Sir) David Cox, logistic regression is a classification model (i.e., binary or categorical variables) for calculating the probability of an event occurring, such as high opioid dosing or prolonged surgical time predicting the probability of a patient incurring a postoperative ileus (POI) [19,22]. Thus, the dependent variable (outcome) is a discrete value. Mathematically, logistic regression utilizes a sigmoidal function (S-curve). Comparatively, linear regression estimates the dependent (outcomes) variable when changes emerge with the independent (continuous) variable itself. Historically, linear regression was the first regression technique developed by Legendre and Gauss in the early 19th century [22]. Similar to linear regression, logistic regression also operates under the assumption of linear relationships between predictor variables. This characteristic places it within the broader category of Generalized Linear Models (GLMs) [22]. Linear regression estimates the dependent variable when changes emerge, with the independent variable accurately reflecting the classical single-predictor case. Linear regression overlooks the broader applicability in modeling relationships between multiple predictors and a dependent variable. Polynomial regression, a specific example of multivariable linear regression, highlights the flexibility of the technique, as it simply requires adding higher-order terms of the original predictor to the model, eliminating the need for specialized software [5,22]. Understanding the generalizability of linear regression beyond single-predictor scenarios is crucial for effectively applying this powerful tool to analyze complex data with multiple influencing factors.

One of the limitations of a linear regression model is that a straight line is forced to match data points for an outcome, where the straight line may not pass through most data points [8,14]. Although linear regression may not pass through most data points, it may still generalize better than polynomial regression, which may overfit the dataset. Logit and probit functions generate S-shaped curves, readily identifying the largest data points on an XY-axis (X: independent variable; Y: dependent variable) (Figure 2) [20].

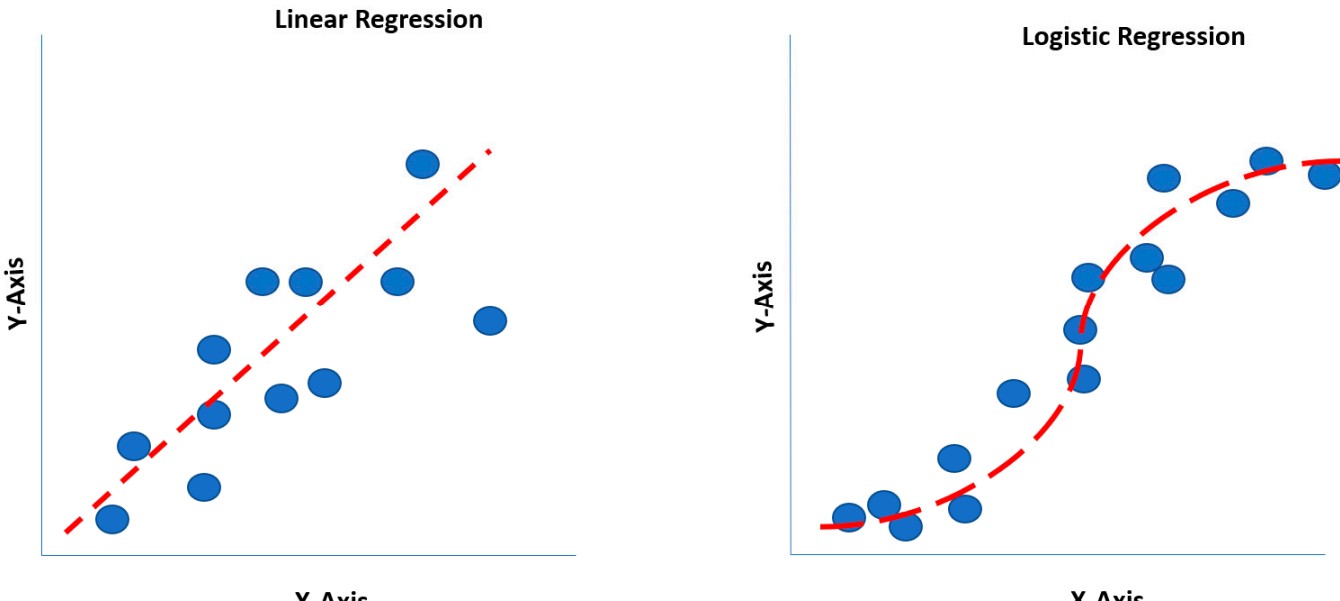

**Figure 2.** Linear regression versus logistic regression comparison.

A mathematical calculation called min log loss or cross-entropy minimization is performed to identify the correct S-curve, enabling the best fit on the data points. The power of logit and probit functions is their multi-dimensionality. Logistic regression benefits include classification models that provide probabilities, cross over to multiple classes (i.e., multinomial regression), and rapidly train and classify unknown (new) data. However,

disadvantages include constructing linear boundaries and assuming that independent variables are independent, and coefficient interpretation remains challenging [8,14,20].

Through logistic regression, classification predictions can be made. A confusion matrix is used to evaluate the classification model after training and validation for accuracy, recall, precision, and F1 scores. A confusion matrix is a quantification of the False Positives and False Negatives, which is well known across the medical field. A confusion matrix is a 2 × 2 matrix summarized by the following (Figure 3) [21].

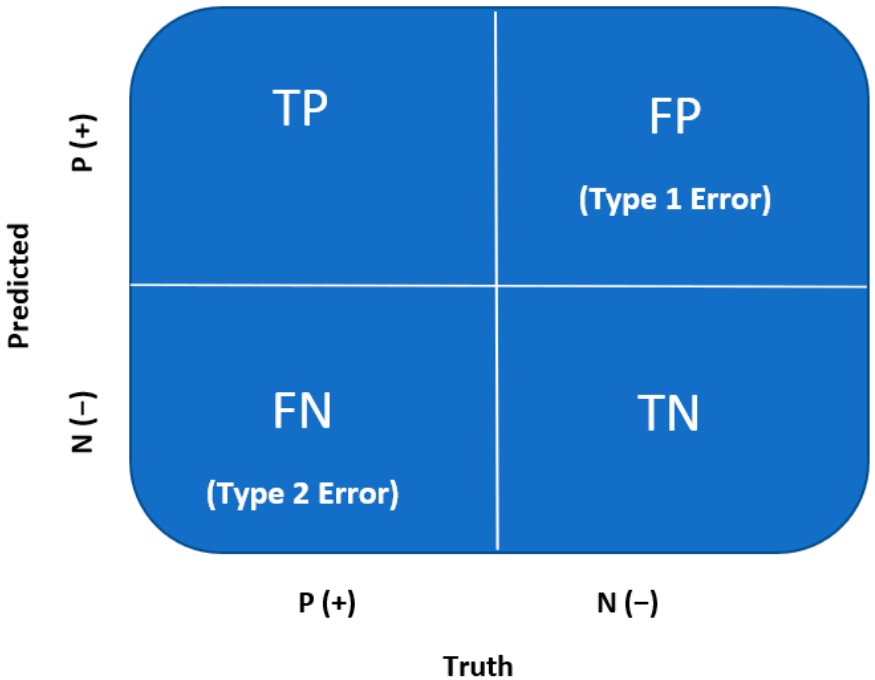

**Figure 3.** Confusion matrix [8,21].

True Positive (TP): Every number predicted true is, in fact, true (positive);
True Negative (TN): Every number predicted negative is, in fact, false (negative);
False Positive (FP or Type I Error): Every number predicted true is, in fact, false (negative);
False Negative (FN or Type II Error): Every number predicted negative is, in fact, true (positive).

## 6. Confusion Matrix

A confusion matrix is a method for evaluating a ML or NN model through performance measures (Table 2) [21].

**Table 2.** Performance measures for confusion matrix [8,21,22].

| Performance Measure | Performance Measure Calculation |
| --- | --- |
| Accuracy | = TP + TN/(TP TN + FP + FN) = 1 − (error rate) |
| Precision | = TP/TP + FP |
| Recall (Sensitivity) | = TP/TP + FN (True Positive Rate) |
| F1 Score | = (2 × Precision × Recall)/(Precision + Recall) |
| Specificity (True Negative Rate) | = TN/TN + FP |
| Classification Error Rate | = Type I Error + Type 2 Error |

TP = True Positive; TN = True Negative; FP = False Positive; FN = False Negative.

Further extrapolation of clinical classification problems can be achieved by utilizing receiver operator characteristic curves (ROC) and thresholds to determine the output

probability (Figure 4) [22]. ROC curves plot False Positive rates against True Positive rates, where the area under the curve measures how well the model performs. The AUC measures how well the model performs (Table 3) [22].

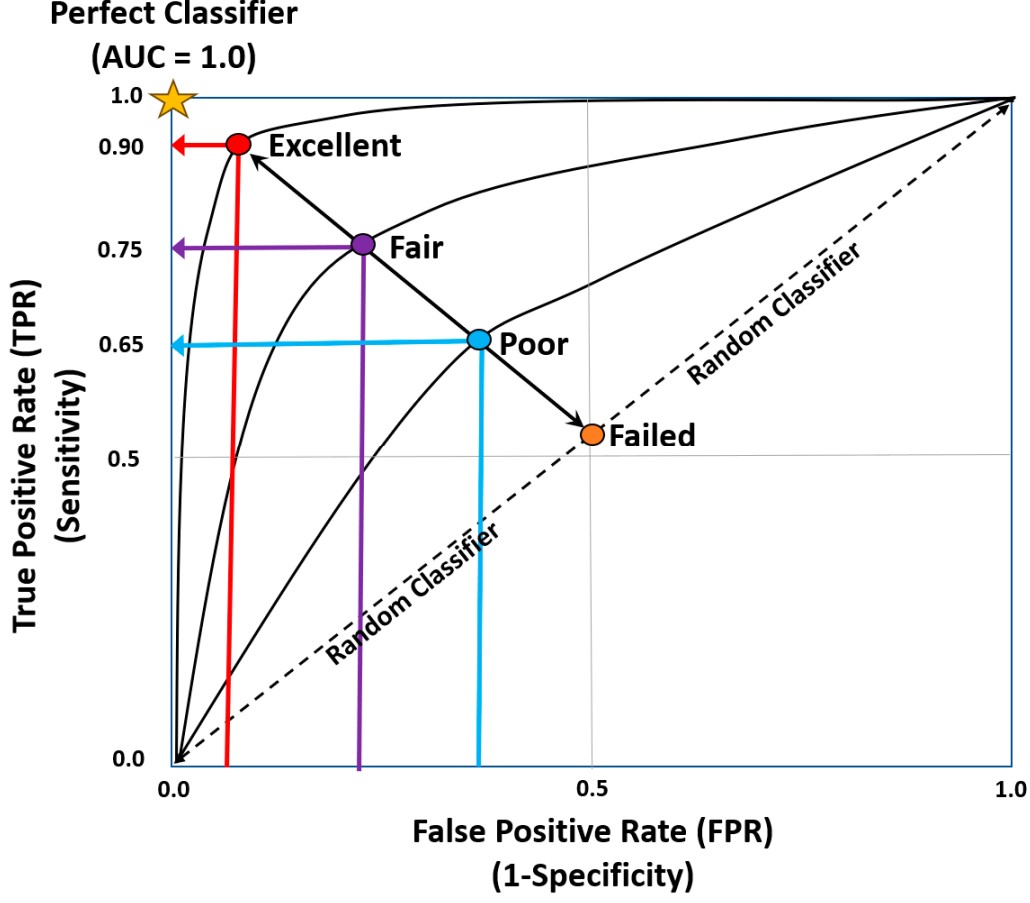

**Figure 4.** Area under the ROC curve [8,21,22].

**Table 3.** Area under the ROC curve values [8,21,23].

| ≤0.5 | Failed |
|---|---|
| 0.5 to 0.7 | Poor |
| 0.7 to 0.8 | Fair |
| 0.8 to 0.9 | Good |
| 0.9 to 1.0 | Excellent |

*Classification (Decision) Trees*

Decision trees are a category of non-parametric models used for classification and regression [8]. Decision trees serve as a predictive modeling tool that facilitates the tracing of various choices (i.e., drug A or B) or solutions to a specific result (i.e., POI). They are applicable to both linear and non-linear datasets, predominantly favoring non-linear ones. Comprising various nodes, the decision tree commences with a root node and concludes at leaf nodes, which represent the culmination of a decision chain or the ultimate result. Beyond the leaf nodes, the decision tree does not extend any further branches. In the context of machine learning, the data attributes are represented by internal nodes, while the result is depicted by the leaf node [8].

Classification (Decision) Trees measure impurity or purity for a given variable grouping. The independent variables emerge from a root node, further splitting down the tree

based on the purity of the groups [8,24,25]. A range of algorithms exist for constructing decision trees based on computational metrics. The metrics determine the similarity of a region or node before splitting to the next branch, which measures the impurity of a region. The larger the regional impurity, the more significant the dissimilarity of the data at that node or region. Some measures of impurity include Gini impurity, entropy, and variance. Some algorithms include Classification and Regression Tree (CART), entropy (C4.5), and Chi-squared Automatic Interaction Detector (CHAID). CART is a popular decision tree algorithm that uses Gini impurity to measure impurity [25]. Gini impurity assesses the frequency of misclassifying an element picked at random from a dataset, assuming it is labeled in a manner consistent with the label distribution of the dataset. CART assigns the decision tree split based on the maximum decrease in Gini impurity called Gini Gain [21,25].

One area for improvement with decision trees is overfitting [25]. However, decision tree and random forest models tend to overfit far less than other ML models. Overfitting occurs when the decision grows fully, fitting perfectly on the training set, but predictions on the test set have reduced accuracy or led to poor generalizability [26]. Overfitting is minimized through pre-pruning or post-pruning techniques. In pre-pruning, a Grid Search cross-validation method utilizes hyperparameter tuning in identifying the optimal hyperparameter values for the ML model. Pre-pruning minimizes decision tree overgrowth through bounding hyperparameters. Post-pruning begins with a full decision tree and then prunes the tree into sequentially smaller trees. The most common post-pruning algorithm is cost complexity pruning. The aim is to identify the "relative error decrease per node" for that given complexity parameter. Cross-validation error is then utilized to determine the best-pruned decision tree [25,26].

While individual decision trees, like the well-established C4.5 algorithm by Ross Quinlan, do not constitute ensemble methods in themselves, they can be effectively employed within ensemble approaches like Leo Breiman's Classification and Regression Trees (CART) technique. This distinction highlights the importance of differentiating between individual learning algorithms and the broader frameworks that combine them for enhanced performance [27–29].

## 7. Bootstrapping in Machine Learning [16,18]

In statistical analysis, the term "bootstrap" refers to a resampling technique where data points are drawn, with replacement, from the original dataset to create new samples of the same size [29]. Each parameter value is assigned a new random value from a probability distribution drawn from its sample space [8]. This process is iterated numerous times, generating a collection of "bootstrap replicates" that approximate the variability within the original data. Bootstrap aggregation refers to bootstrapping (i.e., resampling) the original dataset with replacement [29]. Software typically implements bootstrapping using random-number generators to ensure the unbiased selection of data points. This randomized approach allows researchers to estimate the sampling distribution of various statistics, such as the mean or standard deviation, and construct confidence intervals around them (Table 4) [29]. Overall, bootstrapping provides a valuable tool for exploring the uncertainty and variability inherent in statistical data, informing robust conclusions, and enhancing the reliability of analyses.

**Table 4.** Benefits to Bootstrapping [29].

| Non-Parametric Nature | Applicable to a Wide Range of Data Types without Requiring Specific Assumptions about Their Distribution. |
| --- | --- |
| Flexibility | Can be used to estimate various statistics and assess their variability. |
| Computational Efficiency | Relatively fast to implement, especially when compared to other resampling techniques. |

### 7.1. Ensemble Techniques: Bagging, Random Forest, and Boosting

Ensemble techniques are a robust ML methodology that combines multiple models to generate more reliable predictions for an outcome variable (Y). This technique, called committee methods, is based on diversity in computational models, providing better predictability [25,30–32]. The ensemble method reduces variance from any given model, and each model incorporated into the ensemble is independent. Parallel or sequential building can be used when implementing this technique, depending on an individual's needs (Figure 5).

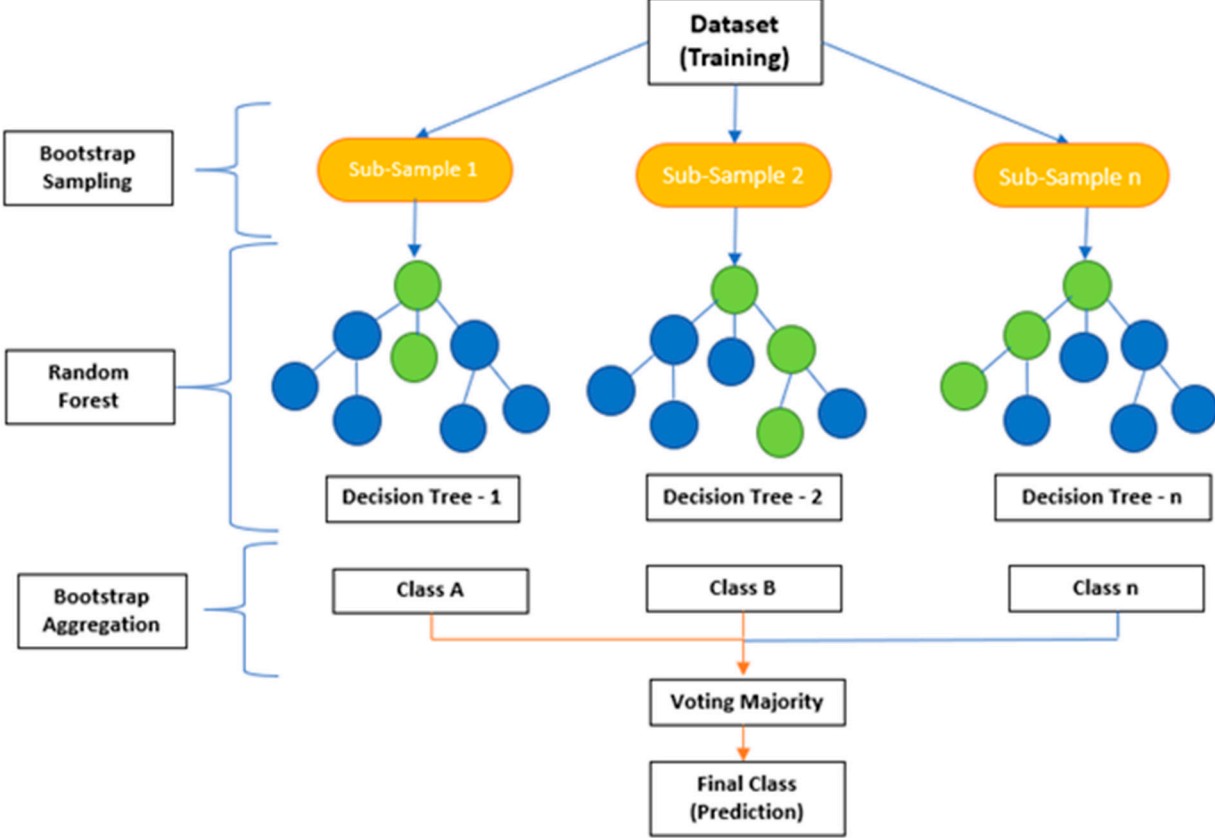

**Figure 5.** Ensemble techniques in machine learning [16,18,31].

Ensemble techniques have become increasingly popular for ML applications because they improve accuracy and reduce variance [21,25]. Ensemble techniques, such as random forest, decision tree classifiers, bagging, and boosting, are supervised learning approaches that can be applied to the understanding of perioperative variables impacting patient outcomes [21]. These methods have become increasingly crucial in perioperative medicine due to the complexity of surgical interventions, anesthetic care, and patient co-morbidities [26]. By leveraging ensemble techniques for predictive analytics, perioperative clinicians can better anticipate potential complications or implement changes intraoperatively or postoperatively to optimize patient outcomes [26,31,32].

Bagging is often used as a form of bootstrap aggregation. Bagging generates diverse models using sampling data with replacement, which is the driver for developing diverse models [29]. The bagging technique builds models in parallel, ranging up to n models in the algorithm [2,8,29].

Decision tree analysis is a valuable ensemble method of analyzing data, but it can take time to interpret its results. A drawback of decision trees is that they tend to overfit, which means that they will perform less well on new data. However, if a collection of many decision trees exists, a technique called random forest can make predictions more stable without overfitting [31–35]. Random forests utilize bootstrap aggregating (bagging)

to combine multiple decision trees, resulting in higher accuracy by reducing bias from overfitting datasets with a single model (Figure 6) [21,25,33–36].

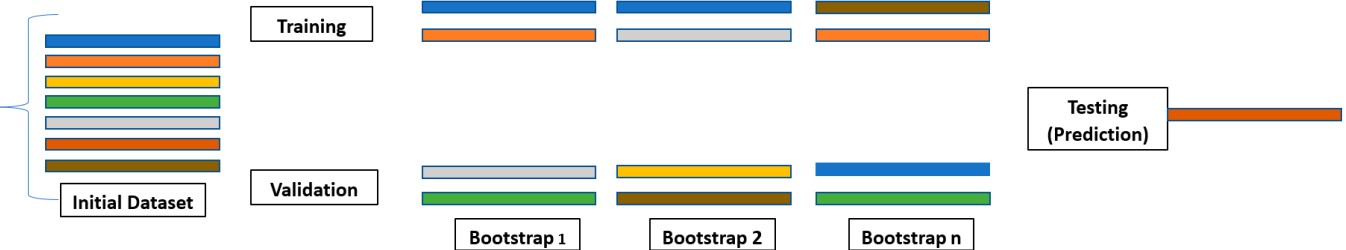

**Figure 6.** Bootstrapping [8,21]. Each color represents a decision tree in the initial dataset. A collection of decision trees allows for a more stable technique to be used called Random Forest. Random forest randomly selects a set of decision trees through bootstrap aggregation known as bagging.

Decision tree classifier algorithms use binary splits through recursive partitioning in datasets. Meanwhile, bagging creates diverse subsets of training samples from larger datasets, reducing variance errors associated with individual models [34,37–40]. Boosting algorithms, on the other hand, combine weak learners into strong ones through iterative processes until the desired model performance is achieved. Compared to traditional modeling approaches like logistic regression or linear discriminant analysis (LDA), boosting algorithms offer improved generalization capabilities [1,2,8,21,25].

The machine learning technique of boosting builds models sequentially, starting with a smaller number of models and increasing the number of models in each iteration [41–45]. The most common boosting methods include adaptive boosting (AdaBoosting), gradient boosting (GBM), and extreme gradient boosting (XG Boost) [40,46,47]. Each successive model is built using the weighted average of previously successful models to reduce error rates on dependent variables while increasing weights on more successful models with lower error rates [48–50].

### 7.2. Supervised Learning Limitations

ML, specifically SL, is a robust mechanism for augmenting correlational determinants of perioperative outcomes. SL and ML have a range of limitations, such as an understanding of SL techniques and interpretation [8,14]. Additionally, criterion-related limitations result from discrepancies between the clinician and the data analyst [14,15]. Another challenge is acquiring a complete set of labeled data that are clean, accurate, and representative of all known outcomes.

### 8. Neural Networks

Neural networks (NN) are interconnected neurons that receive input from various sources such as patient records, laboratory tests, imaging studies, vital signs, and other factors related to pre-, intra-, or postoperative care [51–53]. The NN then analyzes this information using mathematical models, which allow them to identify trends or correlations among different variables so that they can be used for predictive modeling purposes, such as predicting surgical complications or determining optimal treatment plans based on past clinical cases with similar characteristics [53,54]. By leveraging AI technologies like NN, anesthesia providers can gain valuable insights into how their patients will respond during and following surgery before a response even occurs (predictive analytics). Real-time predictive analytics has the potential to drastically improve the quality of care provided by surgeons and anesthesia providers today [55–57]. However, NNs are a growing and diverse set of algorithms that are beyond the scope of discussion in this paper. The use of NNs within anesthesia is still relatively new but offers immense potential for improving patient safety during operations and reducing costs associated with unnecessary interventions. Traditional research methodologies have resulted in changes in medical practice despite

statistically insignificant and clinically unmeaningful results and vice versa [55]. As AI evolves, these applications will continue to play larger roles in helping clinicians to manage clinical practice more efficiently [47,53,55].

## 9. Discussion

### 9.1. Application to the Perioperative Setting (Key Finding)

Supervised machine learning has become increasingly important in surgery, anesthesia, and perioperative care. Ensemble techniques, such as random forests and boosting, combine multiple models for improved prediction accuracy [41–48]. Neural network techniques are also being applied in anesthesiology and perioperative medicine to predict outcomes [51,53,56,57]. ML technology can help to allocate resources more efficiently across the perioperative continuum [58]. Due to the volume and complexity of data, successfully incorporating ML or NNs requires a structural framework. Three trajectories, data science, clinical, and research, comprise the core framework for developing an outcomes program (Figure 1).

The clinical trajectory is the data resource input spanning across several data repositories. The abridgment of the data science trajectory is crucial and instrumental to any form of data analytics and outcomes reporting. Data sources feed a range of data that require standardization into data pipelines before selecting and analyzing any clinical features of interest. After selecting clinical features, predictions, and target outcome variables, model development begins training, validating, and testing across a range of ML models, as described above. Once model development ensues, the research trajectory intersects with the data science trajectory to mature the ML models, develop clinical trials, and implement outcome reporting. The benefit of ML and NN models is the ability to automate the processes in real time through an application programming interface (API). As a result, clinical features of importance can be reviewed in real time for relative impact on a given outcome variable through dashboard applications.

The PCOG researchers developed a triad trajectory framework to explore perioperative oncological outcomes (See Figure 1). The data science trajectory focuses on data validation, cleaning, standardizing, and curation. For example, the data science trajectory emerges from well-developed oncologic data centers housing semi-structured and structured datasets focusing on oncological research, epidemiology, treatment, and patient care. These databases contain large volumes of data that can be easily managed and analyzed with ML and NN algorithms to explore a range of outcome predictions (See Table 5 for data science trajectory resources).

The clinical trajectory in the PCOG conceptual framework is partitioned into five sub-domains, including preoperative, intraoperative, postoperative, post-discharge, and surveillance. Each sub-domain is a reservoir of variables for predicting oncological outcomes. In the preoperative sub-domain, Kowadlo et al. demonstrated the potential of ML algorithms, exemplified by the Patient Optimizer (POP), in accurately predicting postoperative outcomes, including complications like kidney failure and the length of hospital stay [59]. These ML tools provide clinicians with a valuable means to precisely assess preoperative risks, enabling effective patient preparation and potentially reducing perioperative complications and mortality rates. Further refinement and validation of these algorithms through larger prospective studies are essential to enhance their predictive accuracy and broaden their applicability to encompass additional specific complications, readmission rates, and mortality risks [59]. Another study by Ashraf Ganjouei et al. addresses the challenge of predicting clinically relevant postoperative pancreatic fistula (CR-POPF) after pancreaticoduodenectomy (PD), an important complication impacting surgical outcomes [60]. By leveraging ML algorithms, specifically XGBoost, and utilizing preoperative data, the study developed a user-friendly risk calculator for CR-POPF prediction [60]. The XGBoost model demonstrated superior performance with an AUC of 0.72 and identified key predictors including non-adenocarcinoma histology, the absence of neoadjuvant chemotherapy, smaller pancreatic duct size, higher BMI, and elevated

preoperative serum creatinine. Overall, this approach offers promise for enhancing clinical decision-making and patient counseling in the preoperative setting for PD [60]. Lung cancer is a leading cause of death globally, highlighting the urgent need for the precise early detection of nodules in radiology. A study by Syed Musthafa, Sankar, Benil, and Rao addresses this need by proposing a novel hybrid ML approach for early lung nodule prognosis [61]. Leveraging advanced techniques such as snake swarm optimization in combination with a bat model (ISSO-B) and chaotic atom search optimization (CASO), along with a hybrid learning-based deep neural network classifier (L-DNN), the approach aims to improve detection accuracy [61]. Evaluation with public datasets demonstrates promising performance in terms of accuracy, sensitivity, specificity, and area under the curve (AUC), suggesting its potential as an effective tool in lung cancer diagnosis [61].

In the intraoperative subdomain, Nwaiwu et al. showcased the efficacy of ML algorithms, such as decision trees, random forests, and neural networks, in predicting postoperative complications among patients undergoing colectomy for colonic neoplasia [62]. In particular, NN models exhibited high accuracy in anticipating outcomes like anastomotic leak, prolonged length of stay, and inpatient mortality, indicating their potential as valuable tools for perioperative risk stratification. While further validation and optimization are warranted, these ML approaches hold promise in improving postoperative outcomes and aiding clinical decision-making [62]. Szrama et al. demonstrated the efficacy of ML, utilizing the Hypotension Prediction Index (HPI) algorithm alongside arterial waveform analysis, in mitigating perioperative hypotension events among patients undergoing major abdominal surgery [63]. Compared to arterial pressure-based cardiac output (APCO) technology, the HPI algorithm notably reduced the incidence and duration of hypotensive episodes, highlighting its potential to enhance patient safety by pre-emptively alerting clinicians of impending hypotension [63]. These findings underscore the importance of integrating ML-based hemodynamic monitoring systems into perioperative care to mitigate the risks associated with intraoperative hypotension and optimize patient outcomes during major surgical procedures [63]. The utilization of ML algorithms holds significant promise in accurately predicting perioperative outcomes, thereby enhancing patient care, optimizing clinical decision-making, and, ultimately, improving surgical outcomes. A study by Xu, Ju, Tong, Zhou, and Yang, investigated the applicability of ML techniques in predicting postoperative recurrence risk in stage IV colorectal cancer patients [64]. Four fundamental ML algorithms—logistic regression, decision tree, GradientBoosting, and lightGBM—were employed for prediction purposes. The study included 999 patients with stage IV colorectal cancer, randomly divided into training and testing groups at an 8:2 ratio [64]. Their results indicated that the GradientBoosting model exhibited the highest AUC value (0.881) in the training group, while logistic regression showed the lowest (0.734) [64]. In the testing group, the GradientBoosting and GBM models outperformed others, with AUC values of 0.734 and 0.761, respectively, and the GradientBoosting model identified chemotherapy, age, LogCEA, CEA, and anesthesia time as the most influential risk factors for tumor recurrence [64]. Overall, the study demonstrates the efficacy of ML algorithms in predicting recurrence risk in stage IV colorectal cancer post-surgery, with the GradientBoosting and GBM models yielding superior performance [64].

In the postoperative and surveillance subdomains, Jeon et al. employed ML techniques, including LR, support vector machine (SVM), RF, and XGBoost, to predict rectal cancer recurrence following curative resection, with SVM yielding the highest area under the curve (AUC) of 0.831 [65]. Significant predictors of recurrence, such as pathologic Tumor stage (pT), concurrent chemoradiotherapy, and pathologic Node stage (pN), underscored the potential of ML models in stratifying patients for enhanced postoperative surveillance [65]. This study emphasizes the utility of advanced analytics in tailoring follow-up care and improving outcomes for rectal cancer patients, particularly those with elevated pT stages requiring intensified monitoring. In patients undergoing surgical resection for colorectal, liver, and pancreatic cancers, postoperative complications pose significant challenges despite low mortality rates [65]. Merath et al. utilized NSQIP data from 2014 to 2016, and

decision tree models were employed to forecast overall and specific complications [66]. The derived algorithm, based on 15,657 patients, exhibited robust predictive performance for various complications, surpassing traditional risk assessment tools such as the ASA and ACS Surgical Risk Calculator [66]. Notably, the algorithm demonstrated high accuracy in predicting specific complications, including stroke, wound dehiscence, and cardiac arrest, underscoring its potential as an effective risk-stratification tool in surgical settings [66]. Postoperative length of stay following cancer surgery serves as a vital metric for resource allocation and offers insights into surgical outcomes and patient recovery. Leveraging ML techniques on EHRs, a retrospective study by Jo et al. aimed to develop a prediction model for prolonged length of hospital stay after cancer surgery [67]. EHR data from 42,751 patients undergoing primary surgery for 17 cancer types were analyzed, encompassing diverse variables ranging from surgical and cancer-specific factors to underlying diseases and social aspects [67]. Employing the XG boosting classifier, multilayer perceptron, and LR models, the study identified predictors for prolonged postoperative stay, with notable performance observed for kidney and bladder cancers [67]. The incorporation of operative time into preoperative models enhanced predictive accuracy, suggesting the potential of machine learning-based approaches to optimize resource utilization in cancer surgery. Current predictive models for readmission risk often lack specificity to surgical patients and rely heavily on administrative data, potentially limiting their accuracy in colorectal surgery contexts. To address this gap, Howell, Lumpkin, and Chaumont aimed to develop a surgery-specific predictive model for 30-day readmission risk in colorectal surgery patients, incorporating administrative, clinical, laboratory, and socioeconomic status (SES) data [68]. Using a retrospective split-sample cohort of 1549 patients discharged from an academic tertiary hospital between 2017 and 2019, a multivariable LR model was constructed, demonstrating superior performance (C = 0.70, 95% CI 0.61–0.73) compared to internationally used readmission risk prediction indices [68]. This tailored approach, leveraging comprehensive data sources, offers enhanced predictive capability and may facilitate targeted interventions to mitigate readmission risk in colorectal surgery patients [68].

Transitioning into ML applications in oncologic perioperative care, recent advancements have underscored the significance of leveraging data science trajectories and clinical domains to enhance perioperative oncological outcomes. The integration of ML techniques within the PCOG framework offers a transformative approach to predict and manage oncological complications across various phases of perioperative care. Notably, the preoperative phase has seen substantial progress, with studies demonstrating the effectiveness of ML algorithms, such as the Patient Optimizer (POP), in accurately forecasting postoperative outcomes. These tools not only enable clinicians to assess preoperative risks with precision but also hold promise in mitigating perioperative complications and mortality rates. Furthermore, studies like those by Merath et al. and Jo et al. highlight the potential of ML in predicting specific complications and optimizing resource allocation in cancer surgery, indicating a paradigm shift towards personalized and data-driven perioperative care strategies [66,67].

In considering the implementation of ML models in clinical practice, it is crucial to evaluate the associated cost-effectiveness and resource implications. While ML holds promise for optimizing perioperative care and improving patient outcomes, its adoption necessitates the careful consideration of financial factors. Implementation costs may include expenses related to acquiring and maintaining hardware and software infrastructure, training personnel, and integrating ML systems into existing clinical workflows. Additionally, ongoing costs may arise from data management, algorithm refinement, and technical support. However, despite these initial investments, ML applications have the potential to yield long-term cost savings by enhancing diagnostic accuracy, streamlining care delivery processes, and reducing adverse outcomes. Moreover, ML-driven decision support tools may facilitate resource allocation and optimization, leading to more the efficient utilization of healthcare resources. Collaborative research efforts and cost-effectiveness analyses

are essential to elucidate the economic impact of integrating ML into perioperative care, informing strategic decision-making and resource allocation in healthcare settings.

**Table 5.** Data science trajectory resources.

| Program | Country | Website Link | Additional Resources |
|---|---|---|---|
| Surveillance, Epidemiology, and End Results (SEER) Program | United States | SEER Program: https://seer.cancer.gov/, accessed on 10 May 2024 | Explore the following resources to potentially find relevant data science trajectory resources: American Association for Cancer Research (AACR) Careers (https://www.aacr.org/, accessed on 10 May 2024) and National Cancer Institute (NCI) Career Development (https://www.cancer.gov/grants-training/training, accessed on 10 May 2024) |
| National Cancer Database (NCDB) | United States | NCDB: https://www.facs.org/quality-programs/cancer/ncdb, accessed on 10 May 2024 | Same as above |
| Stanford Cancer Institute Research Database (SCIRDB) | United States | SCIRDB: https://med.stanford.edu/ric/data-coordination/scirdb.html, accessed on 10 May 2024 | Same as above |
| Cancer Data Registry of Idaho (CDRI) | United States | CDRI: https://www.idcancer.org/, accessed on 10 May 2024 | Same as above |
| National Cancer Institute of Canada (NCIC) | Canada | NCIC: https://www.cancer.ca/, accessed on 10 May 2024 | Explore the Canadian Cancer Research Society (CCRS) Training and Education: https://www.reproductivefreedomca.org/, accessed on 10 May 2024 |
| Cancer Research UK (CRUK) | United Kingdom | CRUK: https://www.cancerresearchuk.org/, accessed on 10 May 2024 | Explore CRUK's Career Development programs: https://www.cancerresearchuk.org/about-us/careers, accessed on 10 May 2024 |
| Danish Cancer Registry (DCR) | Denmark | DCR: https://ncrr.au.dk/danish-registers/the-danish-cancer-register, accessed on 10 May 2024 | Consider searching for resources offered by universities in Denmark with data science programs |
| Netherlands Cancer Registry (NCR) | Netherlands | NCR: https://www.iknl.nl/, accessed on 10 May 2024 | Explore the Netherlands Organization for Scientific Research's (NWO) career development opportunities: https://www.nwo.nl/, accessed on 10 May 2024 |
| Cancer Registry of Norway (CRN) | Norway | CRN: https://www.kreftregisteret.no/, accessed on 10 May 2024 | Investigate resources at the University of Oslo or other Norwegian universities with data science programs |
| Australian Cancer Database (ACD) | Australia | ACD: https://www.aihw.gov.au/about-our-data/our-data-collections/australian-cancer-database, accessed on 10 May 2024 | Explore resources provided by the Australian Institute of Health and Welfare (AIHW): https://www.aihw.gov.au/reports/workforce/health-workforce, accessed on 10 May 2024 |
| Japan Cancer Surveillance Research Group (JCSRG) | Japan | JCSRG: https://www.ncc.go.jp/en/cis/divisions/stat/index.html, accessed on 10 May 2024 | Consider searching for data science programs at Japanese universities and research institutions |

*9.2. Limitations*

Expanding on the limitations of ML models within perioperative care is essential to gain a comprehensive understanding of their applicability. Despite their potential benefits, ML models are subject to several limitations, including algorithmic biases, model interpretability challenges, and the risk of overfitting to training data. Algorithmic biases may arise from imbalanced datasets or inherent biases present in healthcare practices, potentially leading to disparities in predictive accuracy across patient populations. Moreover, the black-box nature of some ML algorithms complicates the interpretation of model decisions, hindering clinicians' ability to trust and understand their outputs. Furthermore, overfitting can occur when models capture noise or idiosyncrasies in training data, compromising their generalizability to new, unseen data. To address these limitations and ensure generalizability, rigorous validation procedures, such as cross-validation and external validation on diverse datasets, are essential. Additionally, ongoing model monitoring, recalibration, and transparency in reporting model performance metrics are critical for maintaining the reliability and generalizability of ML applications in perioperative care. Collaborative efforts between clinicians, data scientists, and healthcare stakeholders are pivotal in navigating these challenges and optimizing the utility of ML models to enhance perioperative outcomes effectively.

As the integration of ML models in healthcare continues to advance, it is imperative to address the ethical considerations and potential biases inherent in their implementation. ML algorithms rely heavily on historical data, which may perpetuate biases present in healthcare practices, such as disparities in diagnosis and treatment across demographic groups. Additionally, the opaque nature of some ML models poses challenges to understanding how decisions are made, raising concerns regarding transparency and accountability in clinical decision-making. To mitigate these risks, ongoing research and development efforts are essential to enhance algorithmic fairness, interpretability, and transparency. Moreover, interdisciplinary collaborations between clinicians, data scientists, ethicists, and policymakers are critical to establish ethical guidelines and regulatory frameworks governing the responsible use of ML in healthcare. Furthermore, continuous monitoring and evaluation of ML models in real-world clinical settings are necessary to identify and address emerging ethical concerns and biases effectively. By fostering a culture of transparency, accountability, and collaboration, healthcare stakeholders can harness the transformative potential of ML while upholding ethical principles and promoting equitable access to high-quality care for all patients.

The generalizability of ML models presents a significant challenge in translating their promise into real-world clinical practice. Models trained on specific datasets may not perform well when applied to diverse patient populations or in different healthcare contexts. These datasets may not accurately reflect the diversity of patients encountered in clinical practice. Furthermore, inherent biases in data collection practices can lead to imbalanced datasets, where certain patient demographics or disease presentations are under-represented. This can result in models that perform well on the specific data that they were trained on but perform poorly when applied to more heterogeneous patient populations. To ensure generalizability and reliable performance across various perioperative scenarios, rigorous validation studies are necessary. These studies should be conducted in real-world settings with diverse patient populations and ideally involve multicenter collaboration. By evaluating model performance in these more generalizable settings, researchers can identify potential biases and limitations, ultimately leading to the development of more robust and generalizable ML models for improved clinical decision-making in oncologic perioperative care.

*9.3. Future Research*

Despite these challenges, the potential benefits of integrating ML into oncologic perioperative care are substantial. ML models hold promise for enabling personalized risk assessment for patients undergoing oncological surgery. By analyzing patient-specific

data, such as demographics, comorbidities, and tumor characteristics, these algorithms can stratify patients based on their individual risk profiles. This information can then be used to tailor perioperative management strategies, potentially improving patient outcomes. ML-based predictive models can play a crucial role in the early detection and intervention of complications. By identifying high-risk patients for postoperative issues like surgical site infections, these algorithms can enable proactive management strategies to prevent complications and improve outcomes.

The integration of ML can also optimize resource allocation in perioperative settings. Decision support tools powered by machine learning can predict patient outcomes and resource utilization patterns. This allows healthcare providers to allocate resources more efficiently, ensuring optimal utilization of operating rooms, hospital beds, and personnel, ultimately enhancing the efficiency of healthcare delivery.

Clinical decision-making can also be significantly enhanced by ML algorithms. These tools can assist clinicians by synthesizing vast amounts of patient data and providing real-time predictive analytics. From preoperative risk assessment to intraoperative monitoring and postoperative care, ML-based decision support systems have the potential to augment clinical judgment, leading to improved patient outcomes and safety.

Finally, ML integration may pave the way for advancements in surgical techniques and perioperative protocols. By analyzing data on surgical outcomes and identifying factors associated with success, ML algorithms can inform the development of innovative surgical approaches and refine existing practices. This has the potential to improve surgical precision and ultimately enhance patient outcomes in oncologic perioperative care. While challenges exist, the potential benefits of machine learning in perioperative oncology are promising. By addressing the issues of algorithmic bias, model interpretability, data quality, and generalizability, as well as focusing on harnessing the potential for personalized risk assessment, early intervention, resource optimization, enhanced clinical decision-making, and surgical innovation, ML has the potential to revolutionize oncologic perioperative care and improve patient outcomes.

## 10. Conclusions

Big data offers immense possibilities for the surgeon and anesthesiology professionals to garner meaningful information across the perioperative continuum in predicting perioperative outcomes [1,3,9,13,15,19,40,60]. Through supervised ML techniques, a range of independent variables can be efficiently evaluated for predicting a given outcome.

In addition, ensemble techniques have been applied with real-time personalized decision support systems for surgery and anesthesia management. Oncology is one area where NNs have been utilized extensively. NNs and ML models can supplement more timely cancer diagnosis and staging with increased accuracy compared to traditional approaches. For example, artificial neural network (ANN) models using bioinformatic methodologies are more sensitive and reliable in detecting optimal markers for colorectal cancer screening compared to traditional approaches. ANN models can detect liver cancer with 96.7% sensitivity and 87.88% specificity [58]. MIT researchers developed Sybil, an AI model for predicting lung cancer from CT (Computed Tomography) screening. The NN model accurately predicted lung cancer at one year with an ROC-AUC of 0.92 [58,69,70]. Colorectal cancer (CRC) histology classification accuracies can be enhanced through convolutional neural networks (CNNs) [69–71]. In anesthesia, drug delivery devices for propofol infusions incorporate deep learning algorithms to maintain the appropriate depth of general anesthesia [70–72]. Similarly, these technologies can be employed within anesthesia practice settings so that clinical teams can receive timely alerts regarding changes observed in patient physiologic parameters throughout a procedure, which may predict or avoid potential complications before they occur.

Applying supervised machine learning technology to healthcare delivery has potential benefits in improving patient outcomes and delivering enhanced value-based healthcare services. For example, its implementation could lead to the earlier detection and identi-

fication of conditions related not only to anesthesia or surgical interventions but also to other areas such as chronic pain management, thus allowing physicians more significant opportunities for early intervention and prevention rather than waiting until symptoms worsen further down the line and require more costly treatments.

In conclusion, supervised machine learning has great promise for enhancing the quality of care across all aspects of perioperative medicine, leading to better health outcomes among patients undergoing surgical procedures and anesthesia care.

**Author Contributions:** V.G. provided expertise on the surgical and perioperative opportunities for artificial intelligence and real-world data science in healthcare and developed the Perioperative Outcomes Group (PCOG) conceptual framework and the application of artificial intelligence. G.B. developed the initial draft of the manuscript, developed the Perioperative Outcomes Group (PCOG) conceptual framework and the application of artificial intelligence, provided expertise on the technical aspects of artificial intelligence and application in medicine, and revised the manuscript. A.U. provided expertise on the surgical and perioperative opportunities for artificial intelligence and real-world data science in healthcare and provided expertise on the technical aspects of artificial intelligence and application in medicine. All authors have read and agreed to the published version of the manuscript.

**Funding:** This review was not funded by any government agency, private company, or non-profit organization.

**Institutional Review Board Statement:** Not applicable.

**Informed Consent Statement:** Not applicable.

**Data Availability Statement:** Publicly available information referenced in the submission and listed below was analyzed in this review.

**Acknowledgments:** We thank Gilles Benoit, 3M St. Paul, Minnesota, for conducting a critical review and providing feedback. The content is solely the responsibility of the authors and does not necessarily represent the official views of 3M.

**Conflicts of Interest:** The authors declare no conflicts of interest.

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
