# Peer review of "Application of Machine Learning in Predicting Perioperative Outcomes in Patients with Cancer: A Narrative Review for Clinicians"

_curroncol, doi:10.3390/curroncol31050207_

Round 1

Reviewer 1 Report

Comments and Suggestions for Authors

Dear Authors, 
this is a really interesting paper that focuses on an open field of research that will potentially be a breakthrough in medicine. As a clinician, I really appreciated your effort to explain and simply clarify some fundamental information of ML and related sciences. I noticed some minor issues:
- in the paragraph "8. Application to the Perioperative Setting" it would be useful and interesting to add and discuss some of the principal papers and researches that focuses on the role of ML in the setting of the review;
- it would be more accurate to specify that this is a narrative review;
- images are fundamental to understand some parts of the text, however some of them are too small and hard to read (fig. 1, fig. 4, and fig. 6).

Reviewer 2 Report

Comments and Suggestions for Authors

This is a comprehensive review of the methods, metrics, and potential applications of AI, ML, and DL in Medicine.  The authors have done a great job presenting available evidence in a comprehensive and comprehensible manner by healthcare providers.  However, my main concern is that the manuscript does not focus on Cancer or Surgical Oncology patients, as the title indicates, but generally on ML and its implications in healthcare.  They could work a bit more on addressing the specificities of these patients.  For example:

1) Mention databases specifically containing data on cancer (Dutch, Danish, NCDB, etc.)

2) Address specific perioperative issues of the surgical oncology patient, i.e. optimal time between neoadjuvant therapy and surgery, postoperative complications (i.e. leaks after low anterior resections in patients who have received upfront CXR, VTE, etc.).

3) Predict which patients will benefit from neoadjuvant therapy and which should undergo upfront surgery

And so forth.

I believe their manuscript would benefit from reducing the part of generally talking about ML and increasing the part of applications of ML to surgicla oncology specifically.

Additionally, in Figure 1 they should add a key explaining all the abbreviations mentioned in the figure.

Thank you for giving me the opportunity to review this article.

Round 2

Reviewer 1 Report

Comments and Suggestions for Authors

All the issues have been addressed, the manuscript is fine.

Author Response

Thank you very much for your time and constructive input during revision # 1. We are grateful.

Reviewer 2 Report

Comments and Suggestions for Authors

My previous comments have been effectively addressed. 

Now, since you have implemented the term “narrative review”, you need to abide by the SANRA guideline for quality assessment of narrative reviews. For your convenience, you can refer to this checklist as well: https://legacyfileshare.elsevier.com/promis_misc/ANDJ%20Narrative%20Review%20Checklist.pd

or this seminal paper by Green et al.:

https://www.ncbi.nlm.nih.gov/pmc/articles/PMC2647067/

Even if narrative reviews are not as structured as systematic reviews and meta-analyses, you still need to describe some search strategy and follow a specific structure.

I hope this helps. Otherwise, I excellent job.

Round 3

Reviewer 2 Report

Comments and Suggestions for Authors

All suggestions and amendments effectively addressed.